# Moisture Adsorption–Desorption Behaviour in Nanocomposite Copolymer Films

**DOI:** 10.3390/polym15142998

**Published:** 2023-07-10

**Authors:** Farah Aqilah Md Zulkiflie, Norazilawati Muhamad Sarih, Nur Awanis Hashim, Mohd Nashrul Mohd Zubir, Shekh Abdullah, Aida Sabrina Mohd Amin

**Affiliations:** 1Department of Chemistry, Faculty of Science, Universiti Malaya, Kuala Lumpur 50603, Malaysia; aqilazulkiflie@um.edu.my (F.A.M.Z.); aidasabrina8@gmail.com (A.S.M.A.); 2Department of Chemical Engineering, Faculty of Engineering, Universiti Malaya, Kuala Lumpur 50603, Malaysia; awanis@um.edu.my; 3Department of Mechanical Engineering, Faculty of Engineering, Universiti Malaya, Kuala Lumpur 50603, Malaysia; nashrul@um.edu.my (M.N.M.Z.); shekhabdullah@um.edu.my (S.A.)

**Keywords:** polymer film, nanocomposite, desiccant, relative humidity, low regeneration temperature, moisture adsorption–desorption

## Abstract

Dehumidifying air via refrigerant cooling method consumes a tremendous amount of energy. Independent humidity control systems using desiccants have been introduced to improve energy efficiency. This research aimed to find an alternative to the commonly used solid desiccant, silica gel, which has weak physical adsorption properties. It also aimed to overcome the limitation of liquid desiccants that may affect indoor air quality and cause corrosion. This study reports on the synthesis of poly(vinyl alcohol-co-acrylic acid), P(VA-AA), through solution polymerisation by hydrolysing poly(vinyl acetate-co-acrylic acid), P(VAc-AA). This viable copolymer was then incorporated with graphene oxide (GO) at different concentrations (0 wt.%, 0.5 wt.%, 2 wt.% and 5 wt.%) to enhance the adsorption–desorption process. The samples were tested for their ability to adsorb moisture at different levels of relative humidity (RH) and their capability to maintain optimum sorption capacity over 10 repeated cycles. The nanocomposite film with 2% GO, P(VA-AA)/GO2, exhibited the highest moisture sorption capacity of 0.2449 g/g for 60–90% RH at 298.15 K, compared to its pristine copolymer, which could only adsorb 0.0150 g/g moisture. The nanocomposite desiccant demonstrated stable cycling stability and superior desorption in the temperature range of 318.15–338.15 K, with up to 88% moisture desorption.

## 1. Introduction

Temperature and humidity control within a conditioned space is critical for a range of applications. Due to the warm climate throughout the year, the demand for air conditioning systems has sky-rocketed among Malaysians. To achieve a more pleasant indoor environment, it is important to control the indoor temperature as well as humidity [1]. Generally, the favourable range of relative humidity (RH) is between 40–60%. This is because the air will feel warmer as humidity increases since sweat cannot evaporate faster to cool our body temperature. Ideal RH can be achieved by dehumidification of air [2].

Conventionally, regulating the humidity in the air is conducted via the refrigerant cooling method in which the moisture is condensed out by cooling the air below its dew point. The air is then reheated to the desired temperature [3]. Despite being used in most household air conditioners, this method greatly relies on electric energy, accounting as much as 50% of total energy consumption [4,5]. On top of that, the use of refrigerant, commonly chlorofluorocarbon (CFC), has raised concern due to its contribution to the depletion of the ozone layer and global warming [6,7]. As a result, industries have begun to look for energy-efficient and ecologically friendly refrigeration systems, which can be conducted by treating sensible and latent loads independently and omitting the use of CFC as refrigerant [2]. This can be achieved via desiccant assisted air-conditioning system where the air will pass through desiccant material before being cooled by the conventional vapour compression system. Further dehumidification will then be conducted using the regenerated desiccant [8]. The main advantage of integrating solid desiccant devices with a traditional cooling system is that it can reduce the consumption of fossil fuels [6].

In obtaining the highest efficiency of the dehumidification system, the desiccant materials play a very important role in which it should possess low regeneration temperature, high adsorption capacity and cycling stability [3]. One of the widely known and used solid desiccants is silica gel. Having a 3D-polymer structure, silica gel has the microporous structure of internal interlocking cavities, leading to a very high moisture adsorption capacity [9]. A study of 10 different types of desiccant materials were conducted and it was found that 3A silica gel showed preferable performance at high humidity and low regeneration temperature. This is attributed to the generous amount of active silanol groups on the surface of the gel [10]. However, due to having weak physical adsorption properties, its adsorption capacity decreases with the increase in adsorption temperature [9]. To overcome the drawbacks of silica gel, new desiccants had been developed such as zeolite, activated alumina and organic polymers, which are excellent in adsorbing water vapour and may be utilised effectively as solid desiccant materials to remove moisture in a variety of industrial applications [11,12,13,14].

Organic polymer is one of the rising materials for desiccant due to its large surface area and outstanding moisture adsorption [3]. In 2016, Chiang et al. discovered that polyacrylic acid showed superior adsorption capacity with total adsorption of 210 g at higher relative humidity when compared to silica gel which could only adsorb a total of 145 g [15]. In another study, the impregnation of sodium chloride (LiCl) in sodium polyacrylate resulted in an impressively high absorption capacity (0.2 g/g) with low regeneration temperature (353.15 K), indicating a good candidate for desiccant cooling application [16]. Another promising candidate is polyvinyl alcohol (PVA) due to its ability to form film and great hydrophilicity. PVA as a desiccant can adsorb moisture from surrounding air at a faster rate as compared to silica gel and activated alumina [6]. Despite its conspicuous ability to adsorb water, polymer alone has weak mechanical strength due to its hydrophilic nature [17]. Hence, the use of fillers is encouraged [18].

Among other fillers, graphene oxide (GO) is prominent for its remarkable mechanical and chemical properties such as high fracture strength, high Young’s modulus and excellent thermal conductivity [19]. Other than having a simple transfer process, GO is also ready for various chemical modifications which is useful in industrial applications [20]. One of its outstanding properties is its high hydrophilicity. This is due to the presence of carboxyl, hydroxyl and epoxide functionalities results in excellent water transport properties. It was found that water adsorption capacity of GO increased to 0.26 g/g as compared to silica gel which could only adsorb 0.08 g/g of water. This is due to the larger proportion of hydrophilic functional groups in GO. From the isotherm in Figure 1, it can also be seen that GO has higher water uptake at all relative pressures as compared to silica gel [21].

In another study, GO exhibited high water vapor permeability and low nitrogen permeability, which makes it a promising material for desiccant application [22]. The use of GO as filler in polymer membranes has also been studied. In the study, nano-fillers (GO and its composite with TiO_2_) were dispersed into polyamide layer to improve water vapour permeance up to 2820 GPU. It was found that the hydrophilicity and water vapour permeance of the nanocomposite increased with increasing concentration of nano-fillers [23].

This work involved the synthesis of poly(vinyl acetate-co acrylic acid) (P(VAc-AA)) via solution polymerisation, followed by hydrolysis to produce poly(vinyl alcohol-co-acrylic acid) (P(VA-AA)). Polyacrylic acid is widely known as a superabsorbent polymer which could absorb water up to 30 times its original volume which is a promising property for our study. However, its complex adsorption mechanism slows down the overall adsorption rate [24]. The incorporation of GO as a filler, forming a composite desiccant material and the final product was expected to enhance water vapor sorption capacity, faster rate of adsorption and lower regeneration temperature [25,26].

## 2. Materials and Methods

### 2.1. Materials

Vinyl acetate (VAc) (98%, purchased from Sigma-Aldrich (Burlington, MA, USA)), acrylic acid (AA) (purchased from Sigma-Aldrich (Burlington, MA, USA)), hydrochloric acid (analytical grade, fuming 37%, purchased from Sigma-Aldrich (Burlington, MA, USA)), methanol (CH_3_OH) (95%, purchased from Chemiz (Selangor, Malaysia)), sodium hydroxide (NaOH) pellets (99%, purchased from R&M Chemicals (Selangor, Malaysia)) was dissolved in methanol to produce sodium hydroxide solution, graphene oxide was prepared in-house by our group members, calcium hydride (93%, purchased from Acros Organics (Geel, Belgium)), benzoyl peroxide (with 25% H_2_O, purchased from Merck (Darmstadt, Germany)) and distilled water. All chemicals were used as received except for vinyl acetate and acrylic acid were purified prior to use.

### 2.2. Polymerisation of Poly(Vinyl Alcohol-co-Acrylic Acid), P(VA-AA)

Prior to this experiment, different ratios of monomers and vinyl alcohol:acrylic acid (5:5, 6.5:3.5, and 6:4) were tested. However, the resulting samples did not meet our expectations as they exhibited the formation of a sticky gel and poorly formed film. Only at a ratio of 7:3, a satisfactory sample was obtained.

The copolymers containing 7:3 ratios of VAc (0.114 mol)/AA (0.066 mol) were prepared by solution copolymerisation in ethanol/water solutions (93/7 *w*/*w*) at 348.15 K using 0.7% (*w*/*w*) benzoyl peroxide (0.454 mmol) based on the mass of the monomers as the initiator in a 250 mL round-bottom flask. Copolymerisation was carried out in solutions containing 70% in volume of solvent and 30% in volume of the VAc/AA comonomer mixture. The reaction mixture was bubbled using an inert gas for approximately 30 min. The copolymerisation was conducted at 348.15 K for 4 h. The solution of the copolymer in ethanol/water was evaporated to remove all the unreacted monomers and excess solvent. P(VAc-AA) was formed (Figure 2). Yield > 90%. FTIR (cm^−^^1^): 1033 (C-O stretching, VAc), 1229 (C-O stretching, AA), 1376 (C-H bending), 1715 (C=O), 2944 (CH_2_). ^1^H NMR (ppm, CD_3_OD, 400 MHz): 1.65–1.85 (m, 2H, -[CH_3_COOCHCH_2_]_x_-[CH_2_-CHCOOH]_y_-), 1.96 (t, 3H, CH_3_COO), 2.30 (s, 1H, CHCOOH), 4.9 (s, 1H, CHCH_3_COO).

To obtain P(VA-AA), the previously obtained copolymer was completely dissolved in methanol at 333.15 K. 40 wt% of NaOH solution (0.020 mol) was added dropwise into the mixture while stirring and left to react for 3 h at the same temperature. Neutralisation with 6M HCl (0.6 mL) was conducted before the copolymer was filtered and washed. The presence of P(VA-AA) can be deduced by the formation of milky white precipitate. The obtained P(VA-AA) was vacuum dried at 313.15 K. The schematic diagram for the preparation of P(VA-AA) from P(VAc-AA) is shown in Figure 3. Yield > 90%. FTIR (cm^−1^): 1055 (C-O stretching, VA), 1167 (C-O stretching, AA), 1376 (C-H bending), 1723 (C=O), 2957 (CH_2_), 3340 (OH). ^1^H NMR (ppm, DMSO-d6, 400 MHz): 1.52–1.74 (d, 2H, -[CH_2_CH(OH)]_x_-[CH_2_-CHCOOH]_y_-), 2.19 (s, 1H, CHCOOH), 3.42 (s, H_2_O from DMSO-d6), 4.03 (s, 1H, CH(OH)), 4.36–4.66 (m, 1H, CH(OH)).

Degree of hydrolysis of the synthesized P(VA-AA) was determined using the standard JIS K6726 and was found to be at 96%.

### 2.3. Preparation of Copolymer/GO Composite Films via Solvent Casting

GO (0.5%) was dispersed in 15 mL of distilled water in an ultrasonic bath for 60 min at room temperature. Copolymer was dissolved in distilled water (200 mL) at 363.15 K. After the copolymer–water solution has cooled to around 333.15 K, the GO aqueous dispersion was poured into the copolymer solution and sonicated for an additional 15 min at room temperature. The composite was then concentrated to be viscous and poured into a Petri dish. The composite was dried in the oven at 333.15 K overnight to form film. The resulting film was named P(VA-AA)/GO0.5. The GO mass ratio was calculated by
(1)$$GO\%=\frac{M_{go}}{M_{c}}\times 100\%$$
where M_go_ is the mass of GO and M_c_ is the mass of the composite.

Similar preparation of making composite films with different ratios (2% and 5%) of GO were following the same procedure. The films were labelled as P(VA-AA)/GO2 and P(VA-AA)/GO5, respectively. The schematic diagram for the preparation of composite samples is shown in Figure 4.

### 2.4. Fourier Transform Infrared (FT-IR)

Chemical structure of the samples was studied using PerkinElmer FT-IR spectrum spectrometer. The scanning was conducted in wave number ranging from 4000 cm^−^^1^ to 400 cm^−^^1^.

### 2.5. Field Emission Spectroscopy (FESEM)

The cross-section and surface morphology of the samples were visualize using Hitachi SU8220 using 1.0–2.0 kV.

### 2.6. Adsorption Isotherms

All samples were placed in a humidity-controlled chamber and moisture was sorbed at 298.15 K [3]. The equilibrium of the sorption process was considered conducted at 5 h and the equilibrium sorption capacity (q_e_) was defined as:(2)$$q_{e}=\frac{m_{e}-m_{0}}{m_{0}},$$
where m_e_ is the mass of the sample after 5 h and m_0_ in the mass of the dry sample.

### 2.7. Adsorption Kinetics

All samples were kept in an oven at 328.15 K until constant mass was reached. The samples were then put into a humidity-controlled chamber and moisture was sorbed at 298.15 K [3]. At regular intervals, the samples were taken out from the chamber and weighed using an electronic balance. The sorption quantity (q) of a sample was defined as
(3)$$q=\frac{m_{t}-m_{0}}{m_{0}},$$
whereby m_t_ represents the mass of the sample at any time.

### 2.8. Desorption Kinetics

Hydrated samples were put in an oven at different temperatures (318.15 K, 328.15 K and 338.15 K) and their weight were recorded at different time intervals.

### 2.9. Cycling Stability

Sorption–desorption cycles were conducted at 298.15 K, 80% RH for 10 times to examine the reusability of the desiccant films. The sorption capacity and sorption rates of the film were recorded to evaluate the stability of the film.

## 3. Results and Discussion

### 3.1. Chemical Structure

In order to study the functional groups present in the copolymer and nanocomposite, the samples were taken into FT-IR analysis. Figure 5 and Figure 6 show the FTIR and ^1^H NMR spectra of P(VAc-AA) and P(VA-AA), respectively.

IR spectrum of P(VA-AA) in Figure 5 (b) reveals an additional broad peak at around 3340 cm^−^^1^, which can be dedicated to the O-H stretching vibrations of the copolymer [27], proving that P(VAc-AA) has been successfully hydrolysed. The peaks at 2957 cm^−^^1^ and 1723 cm^−^^1^ are related to the presence of -CH_2_ and C=O groups from PAA, respectively. Additionally, the peaks at 1376 cm^−^^1^, 1167 cm^−^^1^ and 1055 cm^−^^1^ represent C-H bending, C-O stretching of carboxylic acid and C-O stretching of alcohol, respectively [28].

The conversion of P(Vac-AA) into P(VA-AA) was also confirmed via ^1^H NMR as shown in Figure 6.

From Figure 6a, peaks (a, b and c) as assigned in the spectrum represent the repeating units of VAc, while peaks (d and e) proved the presence of AA repeating units in the copolymer [29,30,31,32]. The absence of sharp peaks (c and b) at around 1.9 and 4.8 ppm, respectively as well as the appearance of new signals ranging from 4.3–4.6 ppm (h–i) in Figure 6b confirms that PVAc has converted into PVA. Peaks at 4.3–4.6 ppm correspond to the hydroxyl proton of PVA in the hydrolysed copolymer [33]. Peaks g, j and k represent the repeating unit of AA and VA, as mentioned in Section 2.2.

Figure 7 shows the comparison of FTIR spectra for P(VA-AA), GO and P(VA-AA)/GO2. From Figure 7b, the IR bands appearing at 1057 cm^−^^1^ and 1382 cm^−^^1^ signify the presence of C-O stretching and vibration in GO, respectively. The broad, medium peaks ranging from 1600–1800 cm^−^^1^ are assigned to sp2-hybridized C=C and C=O stretching, as well as O-H bending of carboxylic acid on the surface of GO [34,35]. Furthermore, a strong broad feature of overlapping vibration modes between 3500 and 2900 cm^−^^1^ can be attributed to O-H groups on the GO surface [36].

The FTIR spectrum of P(VA-AA)/GO2 nanocomposite in Figure 7c exhibits the characteristics of the GO band along with P(VA-AA). Major peaks of GO and copolymer are preserved with slight changes in intensity and position. For example, the C=O peak from P(VA-AA) shifted to 1712 cm^−^^1^ which indicates the interaction between the hydroxyl group of PVA and the carboxyl group from PAA [27]. The existence of hydrogen-bonding interactions between the OH- bonds may be a plausible explanation for these alterations, indicating that in P(VA-AA)/GO nanocomposite, the primary interaction between the raw materials comes from the OH- groups that are depicted in Figure 4 [37].

### 3.2. Surface Morphology

The surface morphology of P(VA-AA) is shown in Figure 8a–d. FESEM images of P(VA-AA) reveal uniform layers indicating that vinyl alcohol (VA) and acrylic acid (AA) form a homogeneous and stable membrane structure. The images also show the porous and uneven structure of the film which are crucial properties for desiccant. This is because higher porosity and surface roughness lead to a bigger surface area, representing higher adsorption capacities [38]. The energy dispersive X-ray (EDX) spectra in Figure 8e shows the presence of 67.13% carbon (C) and 32.87% oxygen (O).

Figure 9a shows the FESEM images of P(VA-AA)/GO2, while Figure 9b represents the images of P(VA-AA)/GO5. From Figure 9a, it was found that GO particles were uniformly dispersed throughout the polymer matrix without any agglomeration. This indicates that the GO particles were homogenously distributed onto P(VA-AA) matrix which was consistent with the result of FTIR. However, when the concentration of GO was increased to 5%, the homogeneous dispersion decreased, and the nanoparticles agglomerated, as seen in Figure 9b. It was reported that at higher loading, more stress is formed, initiating the agglomeration of the nanoparticles [39].

### 3.3. Contact Angle Measurements

Contact angle measurement is an important parameter to determine the hydrophilicity of the prepared samples. In line with previous studies, the contact angle of P(VA-AA) was 51.5° [40,41]. Figure 10 shows the adsorption of water when in contact with the films from 0 to 5 min. It can be observed by the naked eye that the thinning of water droplet in contact with the films as time elapsed. This indicates that there was a very fast adsorption rate of each film.

Table 1 depicts that the contact angle of the nanocomposite film exhibited a decrease in value of up to 31% as compared to the pristine copolymer. The reduction in contact angle is attributed by the abundant presence of hydroxyl and carbonyl groups on the GO surface which enhanced their interaction with water molecules via hydrogen bonding [42]. From Table 1, it can also be observed that the contact angle of P(VA-AA)/GO2 remained constant after 4 min. This is because the film has started to swell, forming bubbles which makes it hard to determine the time at which the water was completely adsorbed. Hence, it was assumed that complete wetting occurs within 3 to 4 min. As for P(VA-AA), the contact angle continued to decrease after 5 min and complete wetting was assumed to occur at around 6 to 7 min, taking slightly longer time than its nanocomposite. The decrease in contact angle and complete wetting time have confirmed the improvement in hydrophilic properties of the nanocomposite films with the loading of GO fillers [43]. Increased hydrophilicity is a desirable feature for desiccant materials because this means that more water can be adsorbed onto the desiccant surface.

### 3.4. Adsorption Isotherms

Surface adsorption and absorption of the polymer matrix make up the overall water vapour sorption stages of polymer films. In the first stage, stable water adsorption on the surface of the desiccant occurred due to hydrogen-bonding between water molecules and the hydroxyl groups of the copolymer when water vapour molecules reach the surface of the desiccant membranes. This stage is followed by the accumulation of water molecules on the surface of the desiccant which causes them to concurrently seep into the polymer matrix due to osmotic pressure [3,44,45].

Figure 11 shows that the adsorption isotherm of both pure copolymer and nanocomposite desiccants have essentially identical shapes, indicating a comparable adsorption mechanism that was not influenced by the addition of GO [46]. Based on IUPAC classification, both materials exhibit type-III isotherms, suggesting that these desiccants are macroporous adsorbents that primarily adsorb water vapour through capillary condensation, in which multilayer adsorption occurs in a porous medium and proceeds to the point at which pore spaces become filled with condensed liquid from the water vapour [47]. Additionally, FESEM images of P(VA-AA) in Figure 8 show the presence of a porous structure, supporting the adsorption of water vapours via the capillary condensation process.

The adsorption capacity for all desiccants increases with increasing RH. For example, the equilibrium adsorption capacity of P(VA-AA)/GO2 increases from 0.007 g/g at 60% RH to 0.245 g/g at 90% RH. This is due to the increase in water vapour in the surrounding which enhances the hydrogen bonding between water molecules with the hydroxyl and carboxyl group of the desiccants, allowing water molecules to penetrate deeper within the polymer structure [14]. At lower RH, the lower adsorption capacity was observed because the pressure of the water molecules was insufficient to penetrate the structure of the desiccant [48]. This proves that the hydrophilic nature of these materials increased with increasing humidity, and they adsorbed a much higher amount of water molecules, which was primarily due to the capillary condensation process involving interconnected capillary channels [46]. The trend of the graph also implies that the desiccant films could easily be generated at lower RH, similarly, as reported for polyacrylic acid sodium salt [49] and N-isopropylacrylamide [14].

Despite being known for having hydrophilic groups that can attract water molecules, pristine copolymer’s polarity is inadequate to achieve a significant adsorption capacity [3]. As observed from Figure 11, the sorption capacity increases drastically with the addition of GO. This finding agrees with the measurement of the contact angle in Table 1, addressing the composite being more hydrophilic. This is due to the presence of oxygen-containing functional groups on GO’s hydrophilic surface and the formation of a hydrogen bond (H-bond) network with water molecules in GO, hence increasing the hygroscopicity of pure copolymer [21,50]. Apart from that, GO is prone to van der Waals forces which causes stacking between the sheets [51]. This random stacking form interlayer spacing that enlarges with the presence of humidity [52]. This phenomenon allows water molecule to reside in the interlayer spacing at increasing humidity, forming H-bond with epoxy and hydroxyl functional groups on GO. As seen in Figure 12, GO platelets are now clearly segregated from one another and interact through a network of H-bonded water molecules rather than directly with one another [53]. To conclude, the increase in water sorption capacity is mainly due to the presence of hydroxyl and epoxy functional groups on GO and H-bond with water molecules. However, the sorption capacity decreases after the percentage of GO surpass 2%. This is contributed by the agglomeration of GO which decreases the surface area of the desiccant, hindering more water molecules to be adsorbed [54]. The agglomeration of GO nanoparticles was confirmed in FESEM images in Figure 9b.

### 3.5. Adsorption Kinetic

The sorption kinetics of the films were examined in this work because it is a crucial metric for assessing a dehumidifier. Dynamic adsorption performance of pure and composite copolymers was tested and compared using a humidity and temperature-controlled chamber. The test condition was set at 298.15 K with various RH%. The adsorption capacity of a solid desiccant is represented as the mass of water adsorbed per mass of desiccant [55].

Figure 13 shows that the adsorption capacity increases sharply during the initial sorption stage, followed by a steady and slow increase approaching saturation. Additionally, the enhancement in water sorption quantity is closely connected to the impregnated graphene oxide, whereby the sorption capacity increases with the addition of GO. The composite desiccants adsorb more water vapour compared to pure copolymer due to the presence of oxygen-containing groups on GO, such as C=O, –OH, C-O, which yield strong acting forces with polar water molecules. Such interactions allow GO to adsorb more water molecules, increasing their sorption capacity [56].

To better explain the dynamic properties of composite desiccants, adsorption rate coefficients were determined. Referring to Figure 13 that shows sorption curves approaching near-exponential shape, the rate of adsorption is calculated by Linear Driving Force model (LDF) [57]. Due to its simplicity and adaptability, the LDF model is frequently used to simulate the adsorption kinetics of gases such as water vapour, nitrogen, oxygen and carbon dioxide [58,59]. The LDF model is expressed as
(4)$$\frac{dx}{dt}=k(x-x_{t}),$$
where dx/dt is the sorption rate, k means the rate coefficient (s^−^^1^), x the equilibrium water sorption capacity (g/g) and x_t_ is the dynamic water sorption capacity (g/g). Integration of (4) gives
(5)$$x_{t}=x(1-e^{-kt}).$$

Rearranging and applying Napierian logarithm to Equation (5), derives
(6)$$-\ln\left(1-\frac{x_{t}}{x}\right)=kt.$$

The LDF model was fitted into the result of 80% RH since the average of RH in Kuala Lumpur is approximately 80% [54]. This can give us a better view for real life application. The rate coefficients, the square of correlation coefficient (R^2^) and the rate of adsorption of all samples are tabulated in Table 2 below.

From Table 2, it is evident that linear relations with good correlation coefficients (R^2^ > 0.98) were achieved, confirming that the water adsorption kinetics obeyed the LDF model.

The obtained k values were then used to calculate the rate of adsorption for each respective sample. The rate of adsorption of all composites surpassed their pure copolymer. Since hydrophilic functional groups on the surface of adsorbents act as primary adsorption components which formed hydrogen bonds with the water molecules, the abundance of hydroxyl and carbonyl groups on the surface of GO played an important role in increasing the rate of adsorption [60].

### 3.6. Cycling Stability

Any adsorbent that is to be utilised in a variety of industrial applications must be capable of being used again for several cycles of adsorption and regeneration [14]. Any desiccant material that is able to maintain its equilibrium sorption capacity not only performs better but also lowers the cost of the entire operation [46].

Based on water adsorption tests, P(VA-AA)/GO2 showed the highest adsorption capacity among other samples. Therefore, its cycling stability was investigated to compare the sorption quantity of moisture at various times throughout the cycling test. The desiccant was hydrated at 80% RH, 298.15 K for 5 h and after complete hydration, the desiccant was regenerated in the oven at 333.15 K to be used in the next adsorption cycle. From Figure 14, it can be observed that the adsorption capacity remained constant during the 10 adsorption–desorption cycles.

To determine the changes of sorption rate after the cycle test, the amount of sorption by the composite during the 150th minute of the sorption process was also recorded. Interestingly, the sorption capacity of the desiccant remained unchanged. These results demonstrate that P(VA-AA)/GO2 exhibits good cycling stability [3].

### 3.7. Desorption

Solid desiccants can become saturated with water and lose their efficacy over time [61] and they must be regenerated to regain their capacity to adsorb moisture. Regeneration is usually conducted by heating whereby water vapour is desorbed from the desiccant surface. In a desiccant-assisted dehumidification system, regeneration is the most energy-intensive process [3]. Conventional desiccant materials such as silica gel and activated alumina required a lot of energy since their regeneration temperature are typically at 373.15–563.15 K [62,63].

In this study, regeneration of the samples was conducted via heating and the desorption capability of the desiccants were evaluated at different temperatures (318.15 K, 328.15 K and 338.15 K). Figure 15 shows the desorption kinetics of P(VA-AA) and P(VA-AA)/GO2 at (a) 318.15 K, (b) 328.15 K and (c) 338.15 K. The equilibrium sorption capacity of both desiccants decreases with increasing regeneration temperature which shows that more water molecules were desorbed at higher temperature. At higher temperature, more energy was gained by the molecules, hence weakening the attractive forces among the molecules [64].

Pristine copolymer and nanocomposite showed rapid desorption at the earlier stage and slowed down gradually towards equilibrium. This is because, at the beginning of the desorption, water molecules were weakly bonded to the surface of the desiccant as well as the neighbouring atoms, making it easier to be removed [65]. However, as the desorption process continued, more molecules were released, reducing the number of adsorbed molecules on the surface. This leads to the remaining molecules being more strongly bonded to the desiccant surface which slowed down the desorption process.

To better understand the amount of water molecules desorbed from the surface, desorption degree for both samples were calculated using the equation below:(7)$$q_{d}=\frac{q_{ea}-q_{ed}}{q_{ea}}\times 100$$
where q_d_ is the desorption degree, q_ea_ is the equilibrium adsorption quantity and q_ed_ is the equilibrium desorption quantity.

The degree of desorption for P(VA-AA) and P(VA-AA)/GO2 is represented by the bar graph in Figure 16. Consistent with the desorption capacity, the degree of desorption increases as the temperature rises. From the graph, P(VA-AA)/GO2 demonstrated superior desorption capabilities, achieving a desorption degree of up to 88% compared to its pure copolymer counterpart, which only managed to desorb 74% of the water molecules at 338.15 K. Even at lower temperatures (318.15 K and 328.15 K), P(VA-AA)/GO2 exhibited desorption rates of up to 80%. This finding confirms that the newly prepared nanocomposite desiccant can be effectively regenerated using renewable energy sources such as solar energy, which can provide temperatures up to 353.15 K [3].

## 4. Conclusions

In this work, P(VA-AA) and its nanocomposite, P(VA-AA)/GO films were successfully prepared using solvent casting. The presence of GO signals in the FTIR spectrum confirmed the incorporation of GO into the polymer matrix. The nanocomposite desiccants demonstrated excellent water adsorption capacity compared to pristine copolymers. The sorption capacity increased from 0.015 g/g for P(VA-AA) to 0.24 g/g for P(VA-AA)/GO2 when exposed to 90% RH at 298.15 K. Moreover, it was found that the optimal loading of GO was 2%, as the adsorption capacity started to decrease when the concentration of GO reached 5%. This decrease could be attributed to the agglomeration of GO at higher concentrations. Hence, utilising only 2% of GO offers the advantage of reducing preparation cost while enhancing adsorption capacity. The nanocomposite desiccant also showed great cycling stability, as the sorption capacity and rate remained unchanged after 10 repeated adsorption–desorption cycles. Apart from that, P(VA-AA)/GO2 exhibited excellent regeneration properties at relatively low temperatures (318.15 K). Overall, this experimental study proved that the addition of GO in the P(VA-AA) matrix indeed improved the water adsorption capacity and regeneration process, thus indicating that P(VA-AA)/GO is promising as a potential desiccant material to be applied in dehumidification systems.

## Figures and Tables

**Figure 1 polymers-15-02998-f001:**
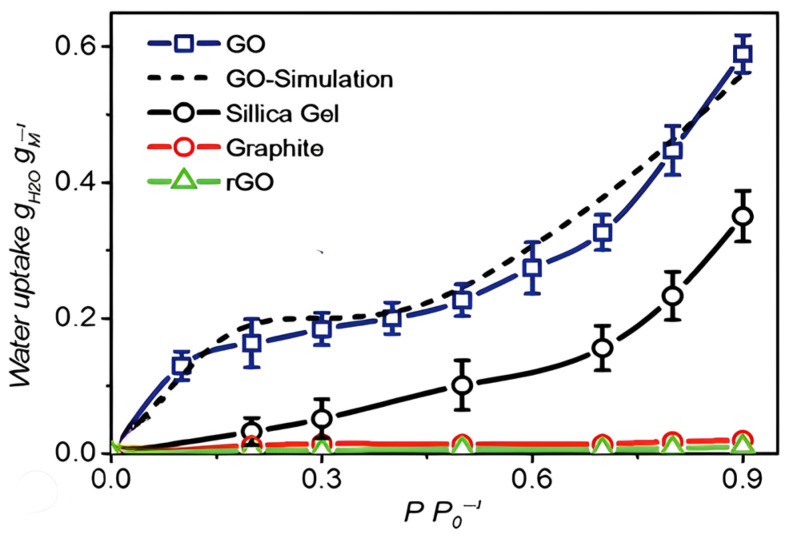
Adsorption isotherms of GO, silica gel, graphite and reduced GO (rGO) at 298.15 K [21]. Adapted with permission from Lian, B.; De Luca, S.; You, Y.; Alwarappan, S.; Yoshimura, M.; Sahajwalla, V.; Smith, S.C.; Leslie, G.; Joshi, R.K. Extraordinary Water Adsorption Characteristics of Graphene Oxide. *Chem. Sci.* 2018, *9*, 5106–5111. https://doi.org/10.1039/c8sc00545a. Copyright 2023, Royal Society of Chemistry.

**Figure 2 polymers-15-02998-f002:**
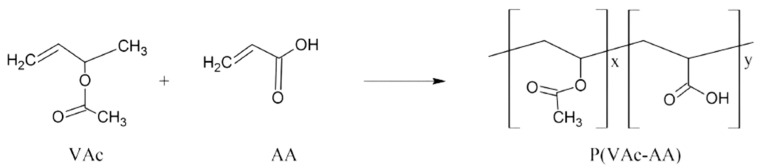
Synthesis of P(VAc-AA).

**Figure 3 polymers-15-02998-f003:**
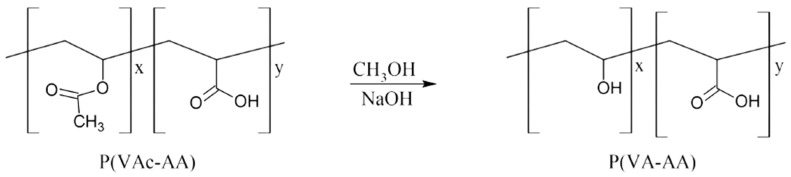
Hydrolysis of P(VAc-AA) to P(VA-AA).

**Figure 4 polymers-15-02998-f004:**
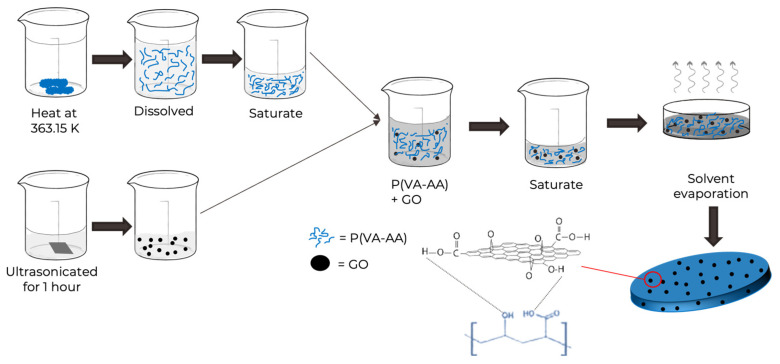
Preparation of P(VA-AA)/GO film via solvent casting.

**Figure 5 polymers-15-02998-f005:**
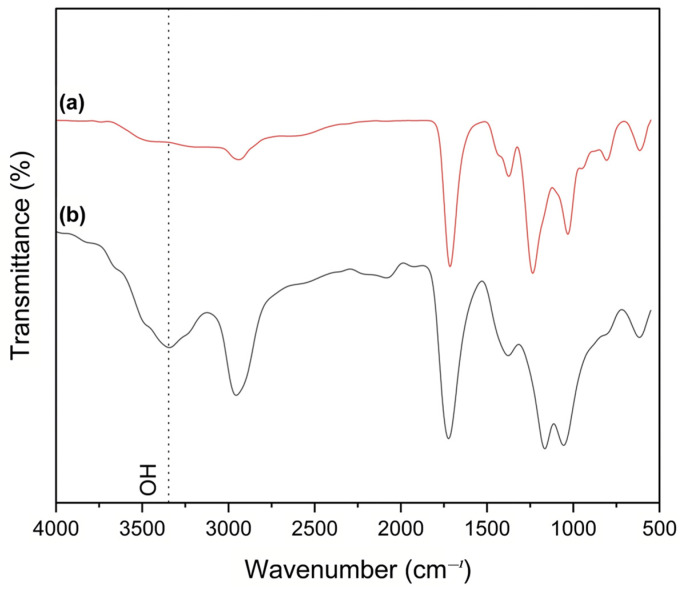
FTIR spectra of (**a**) P(VAc-AA) and (**b**) P(VA-AA).

**Figure 6 polymers-15-02998-f006:**
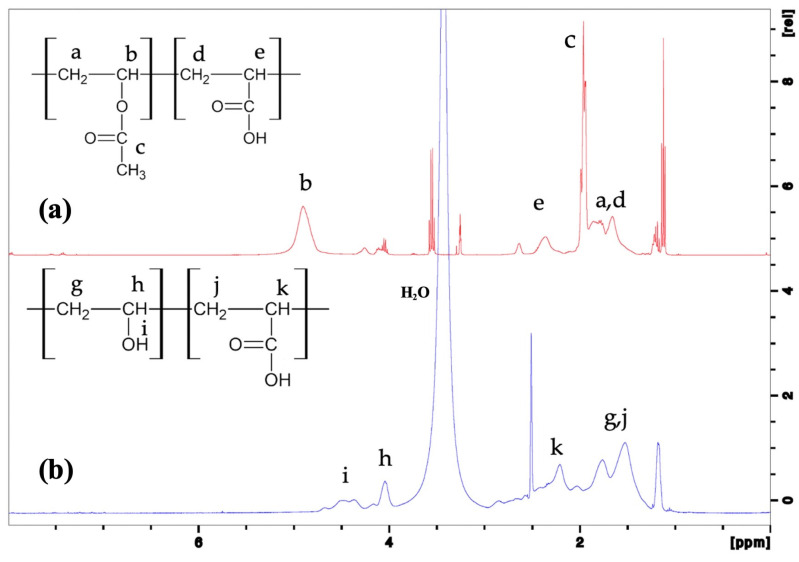
^1^H NMR spectra of (**a**) P(VAc-AA) and (**b**) P(VA-AA).

**Figure 7 polymers-15-02998-f007:**
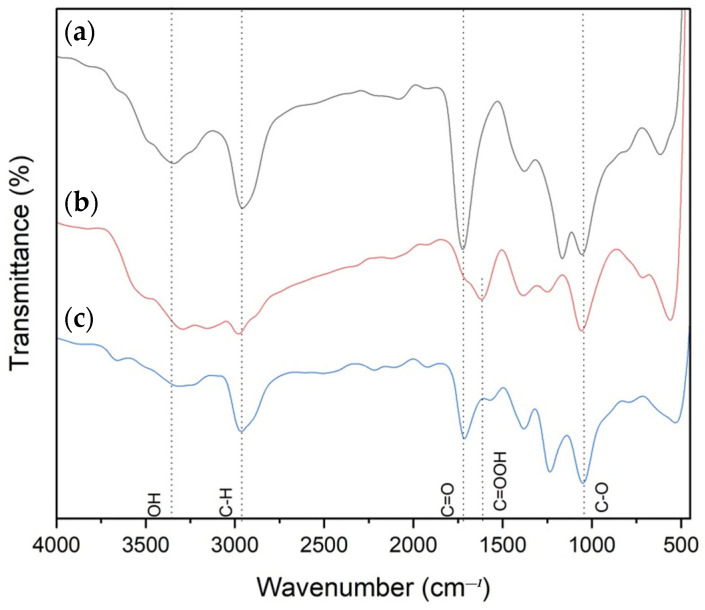
FTIR spectra of (**a**) P(VA-AA), (**b**) GO and (**c**) P(VA-AA)/GO2.

**Figure 8 polymers-15-02998-f008:**
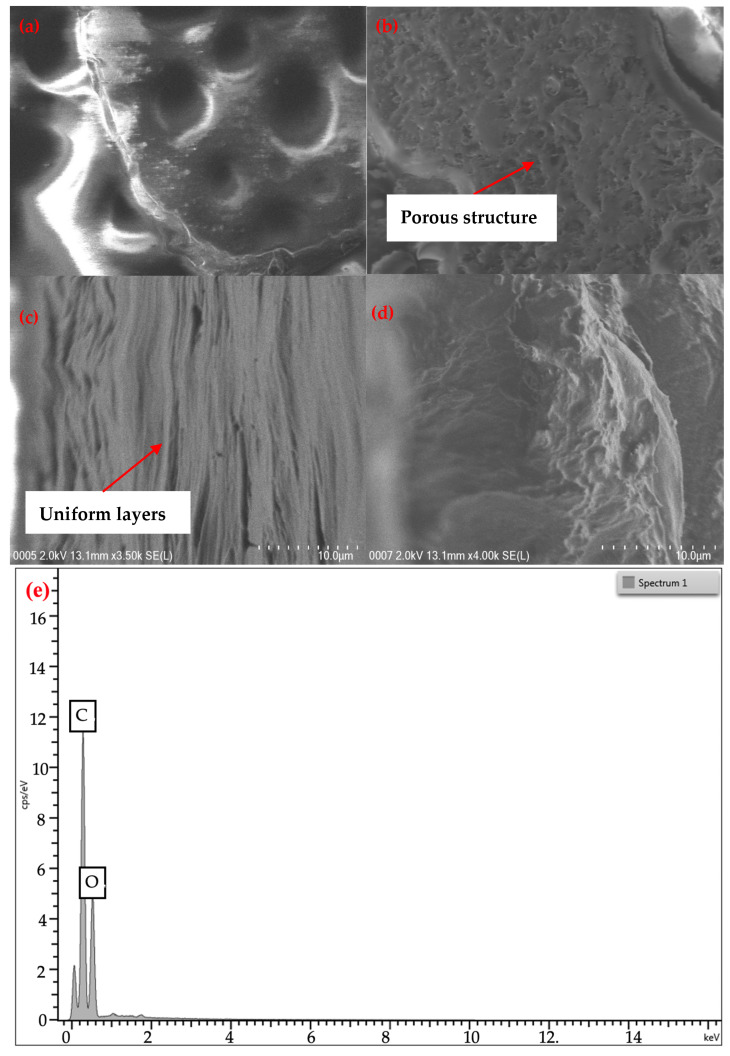
FESEM images of P(VA-AA) at different angles (**a**,**b**) top, (**c**,**d**) cross-section and (**e**) EDX spectra.

**Figure 9 polymers-15-02998-f009:**
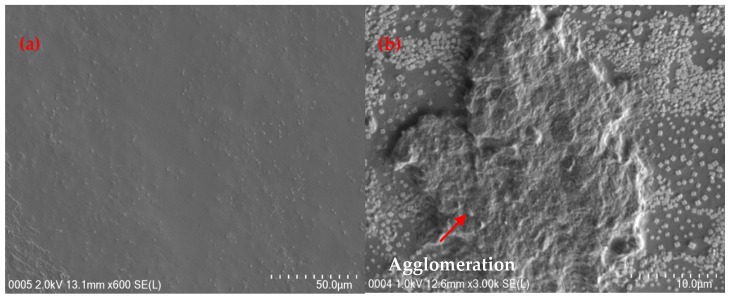
FESEM images of (**a**) P(VA-AA)/GO2 and (**b**) P(VA-AA)/GO5.

**Figure 10 polymers-15-02998-f010:**
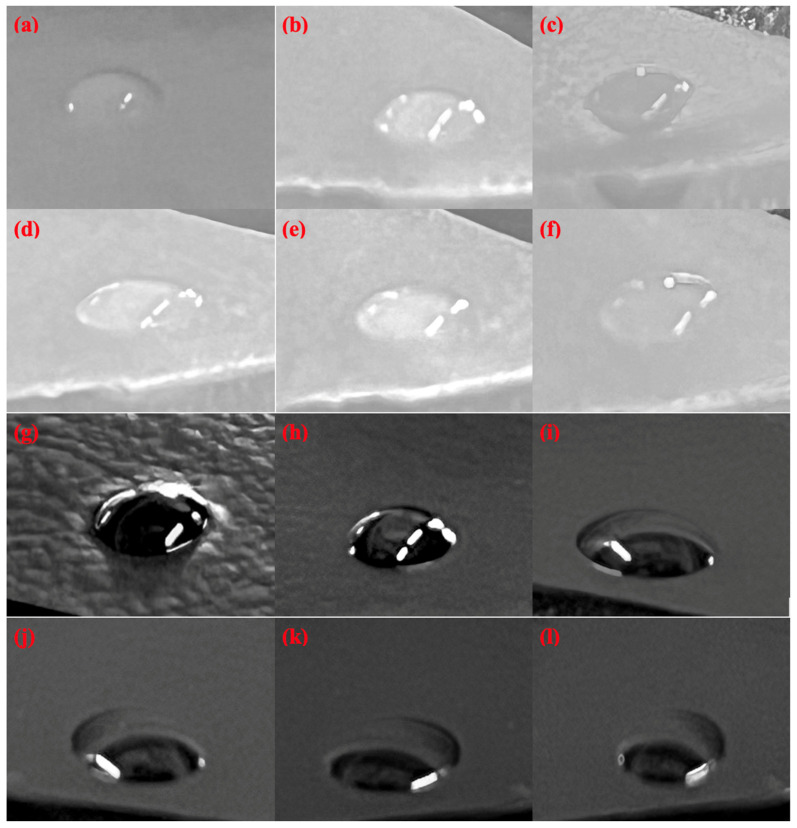
Water droplet images on the surfaces of P(VA-AA) at (**a**) 0 min; (**b**) 1 min; (**c**) 2 min; (**d**) 3 min; (**e**) 4 min; (**f**) 5 min and P(VA-AA)/GO2 at (**g**) 0 min; (**h**) 1 min; (**i**) 2 min; (**j**) 3 min; (**k**) 4 min; (**l**) 5 min.

**Figure 11 polymers-15-02998-f011:**
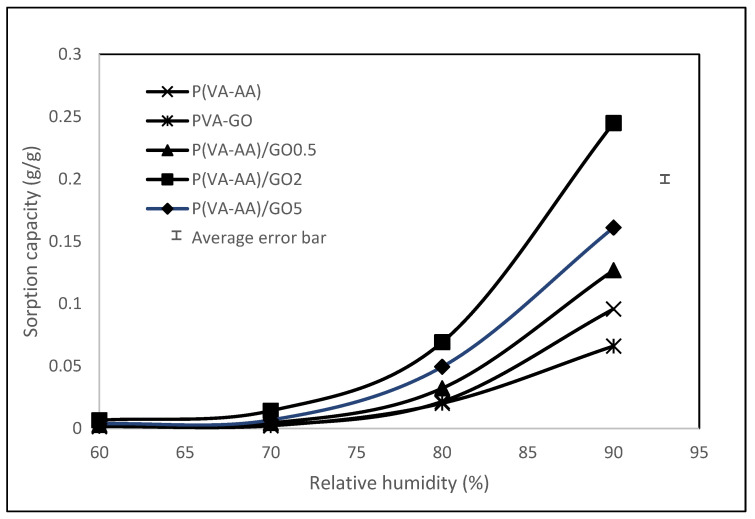
Sorption Isotherm of P(VA-AA) and its composites at 298.15 K.

**Figure 12 polymers-15-02998-f012:**
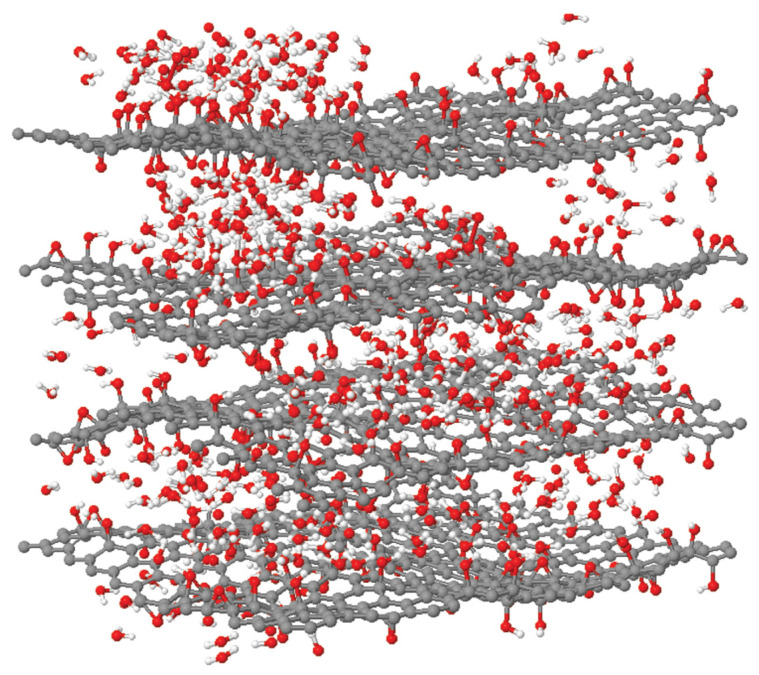
Atomic structure of hydrated multilayer GO [53]. Adapted with permission from Medhekar, N.V.; Ramasubramaniam, A.; Ruoff, R.S.; Shenoy, V.B. Hydrogen Bond Networks in Graphene Oxide Composite Paper: Structure and Mechanical Properties. ACS Nano 2010, 4, 2300–2306. https://doi.org/10.1021/nn901934u. Copyright 2023, American Chemical Society.

**Figure 13 polymers-15-02998-f013:**
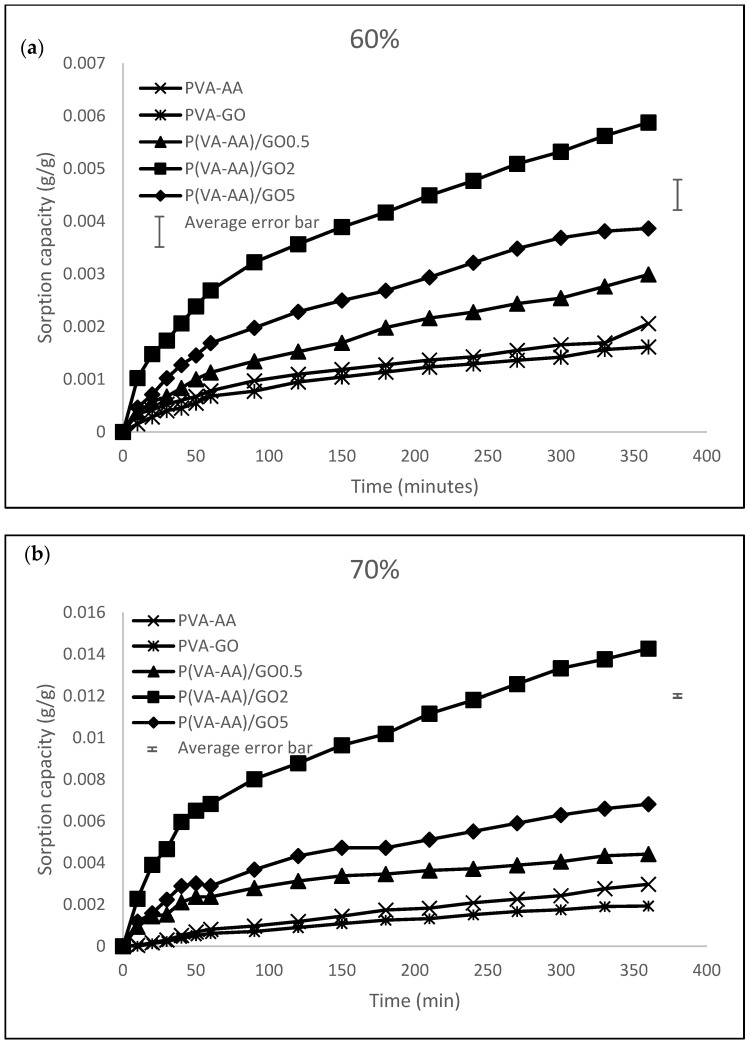
Sorption kinetics of P(VA-AA) and its composites at 298.15 K, (**a**) 60% RH, (**b**) 70% RH, (**c**) 80% RH and (**d**) 90% RH.

**Figure 14 polymers-15-02998-f014:**
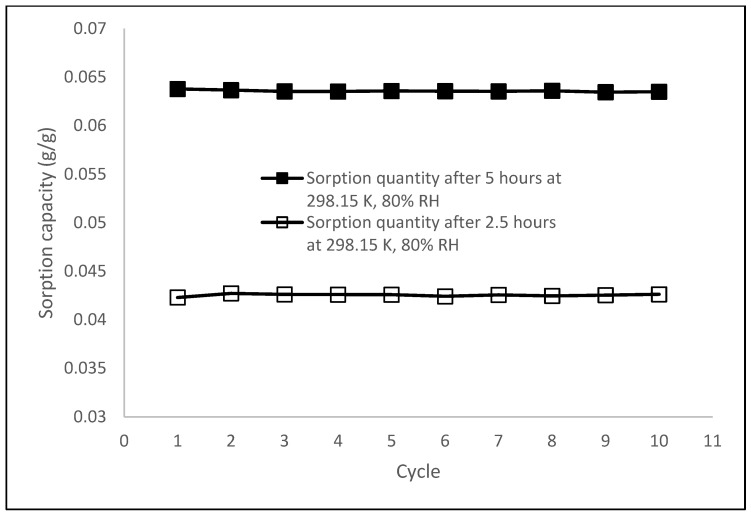
Cycling stability of P(VA-AA)/GO2.

**Figure 15 polymers-15-02998-f015:**
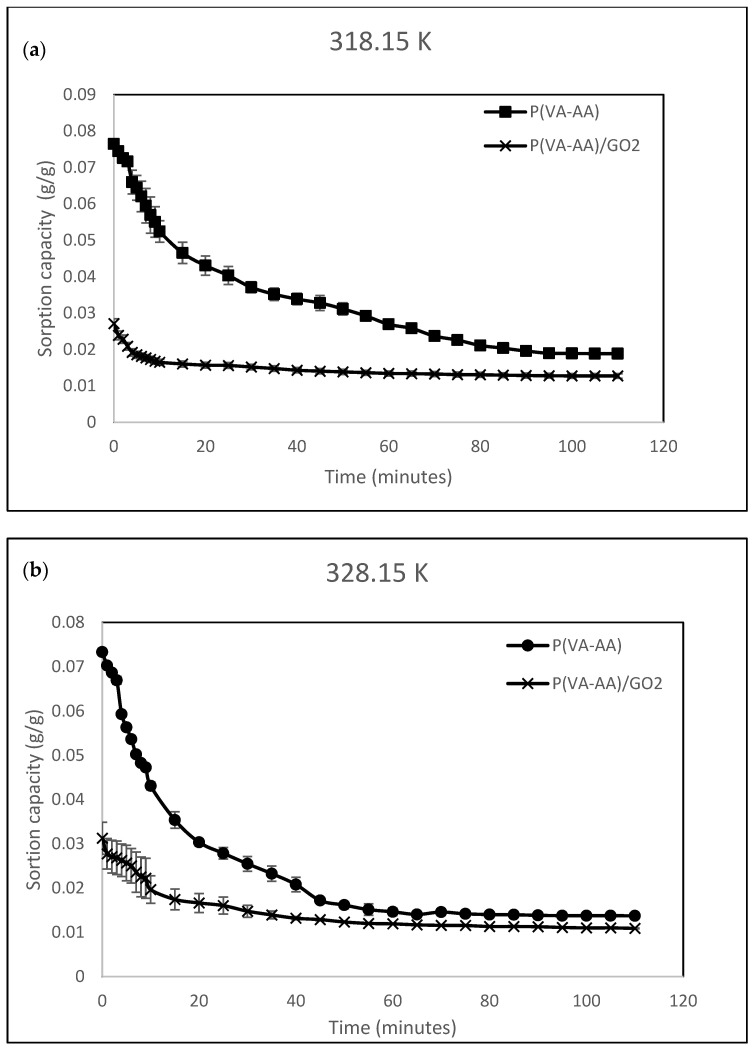
Desorption kinetics of P(VA-AA) and P(VA-AA)/GO2 at, (**a**) 318.15 K, (**b**) 328.15 K and (**c**) 338.15 K.

**Figure 16 polymers-15-02998-f016:**
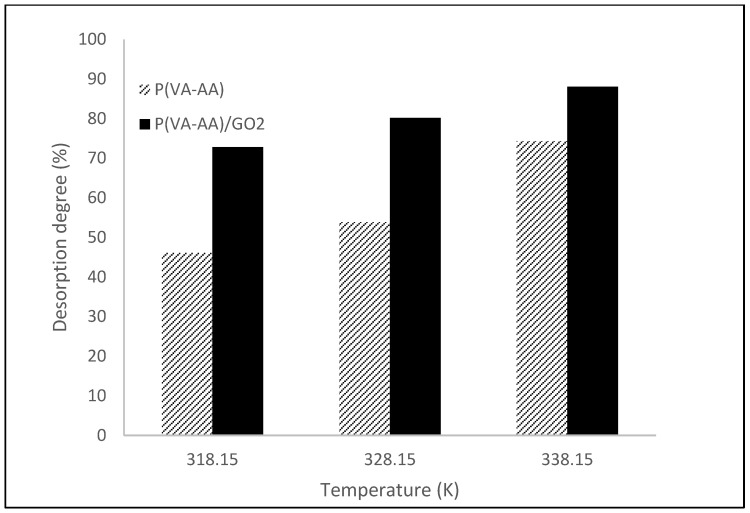
Desorption degree of P(VA-AA) and P(VA-AA)/GO2.

**Table 1 polymers-15-02998-t001:** Static contact angle measurements of P(VA-AA) and P(VA-AA)/GO2.

Sample	Time (min)	Contact Angle (°)
P(VA-AA)	0	51.5
	1	45.5
	2	37.5
	3	32.0
	4	28.5
	5	24.5
	6	20.0
	7	20.0
P(VA-AA)/GO2	0	38.5
	1	29.0
	2	21.5
	3	19.5
	4	17.0
	5	17.0

**Table 2 polymers-15-02998-t002:** Rate coefficients (k), R^2^ and of rate of adsorption (dx/dt) desiccants at 298.15 K and 80% RH.

Sample	k	R^2^	$\frac{dx}{dt}$
PVA/GO	0.0081	0.9896	4.4809 × 10^−05^
P(VA-AA)	0.0083	0.9956	4.4652 × 10^−05^
P(VA-AA)/GO0.5	0.0105	0.9904	6.6953 × 10^−05^
P(VA-AA)/GO2	0.0093	0.9911	1.4528 × 10^−04^
P(VA-AA)/GO5	0.0115	0.9809	1.0802 × 10^−04^

## Data Availability

The author declares that all the data in the article are true and valid.

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
