# Peer review of "Moisture Adsorption–Desorption Behaviour in Nanocomposite Copolymer Films"

_polymers, 2023, doi:10.3390/polym15142998_

Round 1

Reviewer 1 Report

1. The introduction should conclude the super absorbent polymers (SAP), which should be compared with the author’s design (copolymer).

2. Why do the authors select 7:3 of the ratio of VAc:AA, how about other ratios?

3. The copolymer of PVAc-co-PAA and the hydrolyzed PVA-co-PAA should be analyzed in detail. What are the polymer degree, molecular weight and distribution? What is the deacetylation or hydrolysis degree of PVAc. And IR or even NMR spectra should be attached. How do the structures affect the properties?

4. Small amount of GO (2%) added cause the moisture sorption capacity at 0.2449 g/g from the 0.015 g/g, which should be explained deeply, what is the mechanism about GO’s effect and more evidence provided here.

5. The chemical structures of VAc and AA were wrong in Line117, P3.

Author Response

Point 1: The introduction should conclude the super absorbent polymers (SAP), which should be compared with the author’s design (copolymer).

Response 1: Thank you for the suggestion. Conclusions have been added in lines 102 to 107.

Point 2: Why do the authors select 7:3 of the ratio of VAc:AA, how about other ratios?

Response 2: Thank you for the question. Before proceeding with 7:3 ratios, some different ratios have been tested such as 5:5, 6.5:3.5 and 6:4. However, the final product did not meet our expectations, i.e., stickiness and poorly ability to form film. We have added this explanation in the manuscript. Kindly refer to lines 119-121.

Point 3: The copolymer of PVAc-co-PAA and the hydrolyzed PVA-co-PAA should be analyzed in detail. What are the polymer degree, molecular weight and distribution? What is the deacetylation or hydrolysis degree of PVAc. And IR or even NMR spectra should be attached. How do the structures affect the properties?

Response 3: Thank you for the questions. Unfortunately, we do not have the access to the gel permeation chromatography, GPC facility to determine the polymer degree, molecular weight and distribution. However, based on our synthesis, the copolymer is expected to have MW 29,000 g/mol. The degree of hydrolysis of PVAc was calculated and was found to be at 96%. This information has been added in the manuscript in Line 144-145.

To confirm the conversion of PVAc to PVA, IR and NMR spectra of both materials were compared and have been added to the manuscript. Kindly refer to Line 192-213.

The presence of hydroxyl group in PVA aids the interaction of the copolymer backbone with GO as shown in Figure (4). Furthermore, the extra OH groups provided by PVA in the copolymer have enhanceds the hydrophilicity of the copolymer as compared to PVAc. Hence aid the hydrogen bonding with water molecules and leads to better adsorption capacity.

Point 4: Small amount of GO (2%) added cause the moisture sorption capacity at 0.2449 g/g from the 0.015 g/g, which should be explained deeply, what is the mechanism about GO’s effect and more evidence provided here.

Response 4: Thank you for the suggestions. More explanations have been added from Line 326 - 340. The increase in moisture sorption capacity is mainly due to the presence of hydroxyl and epoxy functional groups on GO and H-bond with water molecules. GO is prone to van der Waals forces which causes stacking between the sheets. This random stacking form interlayer spacing that enlarges with the presence of humidity. This phenomenon allows water molecule to reside in the interlayer spacing at increasing humidity, forming H-bond with epoxy and hydroxyl functional groups on GO.

Point 5: The chemical structures of VAc and AA were wrong in Line117, P3.

Response 5: Thank you for the comment. Amendment has been done.

Please refer the attachment. Thank you.

Reviewer 2 Report

Title: ‚Investigation of ‘ not needed.

L. 20: wt.-%?

L. 53-64: Can silica age?

L 62-64: What is new about them and are these better?

L 66-69: Provide values.

L 72: What PVA, different types are available.

L 76-90: Provide values and maybe a water vapour sorption isotherm.

Fig. 1 and Fig. 2: Can a reference be added?

Fig. 5 and Fig. 6: Maybe add arrows and a legend showing what can be seen.

Fig. 8: Add temperature. How to be sure equilibrium was reached? Pure desiccant is missing! When only at hight RH water is absorbed how can it be than a good desiccant?

Eq. 4 to 6: Why equations based on Fick were used such as described by Crank. Than diffusion coefficients could be derived.

L. 358: What humidity? Isotherm for 333.15 K could be provided and compared with 298.15 K.

KL. 384: That happens also at silica gel.

Fig. 11: A comparison with silica gel would be of value. Can this experiment be added?

Author Response

Point 1: Title: ‚Investigation of ‘ not needed.

Response 1: Thank you for the suggestion. We have removed “Investigation of” from our title.

Point 2: L. 20: wt.-%? 

Response 2: We have added wt.% in our manuscript.

Point 3: L. 53-64: Can silica age?

Response 3: Thank you for the question. Silica gel can absorb moisture and lose its effectiveness over time. This is due to the fact that silica gel has a limited capacity for absorbing moisture, and once it has hit its limit, it will no longer be able to absorb any further moisture.

Furthermore, silica gel can get polluted with other elements such as dust or oils, affecting its capacity to absorb moisture. To retain its efficacy, silica gel should be stored in a clean and dry environment.

However, silica gel does not "age" in the same manner that other materials do. Silica gel may be utilised for a prolonged amount of time if properly preserved.

Point 4: L 62-64: What is new about them and are these better?

Response 4: Thank you for the question. Although silica gel is widely used for desiccant, silica gel is fragile and brittle. During a tensile test, silica gel exhibited brittle fracture before yielding behaviour took place [refer https://doi.org/10.1002/(sici)1097-4628(19991003)74:1%3C133::aid-app16%3E3.0.co;2-n & https://doi.org/10.1016/j.enbuild.2015.05.009 ]. Zeolite, activated alumina and polymers as mentioned in the manuscript, are the new alternatives to silica gel desiccant which are excellent in adsorbing water vapour and may be utilised effectively as solid desiccant materials to remove moisture in a variety of industrial applications due to better mechanical properties.

Point 5: L 66-69: Provide values.

Response 5: Thank you for the comment. Values have been added in the corrected manuscript from Line 69 to 72.

Point 6: L 72: What PVA, different types are available.

Response 6: Thank you for your question. PVA referred in the paper is in general form.

Point 7: L 76-90: Provide values and maybe a water vapour sorption isotherm.

Response 7: Thank you for your comment. Values and isotherm have been added in Line 86 to 92.

Point 8: Fig. 1 and Fig. 2: Can a reference be added?

Response 8: Thank you for the question. These figures are general copolymerisation reaction and sketched by the authors. Hence, no reference should be added.

Point 9: Fig. 5 and Fig. 6: Maybe add arrows and a legend showing what can be seen.

Response 9: Thank you for the suggestion. Amendments have been done and can be referred in Line 240 and Line 253.

Point 10: Fig. 8: Add temperature. How to be sure equilibrium was reached? Pure desiccant is missing! When only at hight RH water is absorbed how can it be than a good desiccant?

Response 10: Thank you for your comment, temperature has been added.

The equilibrium was assumed to have reached after 5 hours as stated in line 171. The data for pure desiccant was already included in Figure 11 (line 295) and represented by (X) in the graph.

The purpose of desiccant in our daily life is to adsorb excess humidity in our surrounding. The favourable range of relative humidity (RH) should be around 40-60%. Therefore, a desiccant is considered good when it can adsorb less humidity at lower RH and adsorb more at higher RH. Apart from that, as shown in Figure 11, the prepared samples do indeed adsorb some water vapour at low RH, only not as significant as higher RH. This is to ensure favourable range of RH is maintained.

Point 11: Eq. 4 to 6: Why equations based on Fick were used such as described by Crank. Than diffusion coefficients could be derived.

Response 11: Thank you for your question. The equations chosen in this paper was based on LDF model that are frequently used to simulate the adsorption kinetics of gases including water vapour, which is the main study in this paper.

Are you suggesting that we use Crank equation? If yes, to calculate the diffusion coefficients, concentration at a certain distance in the medium must be determined, in which we do not have the access to. However, since our main topic is the adsorption behaviour of water vapour, we believe that the value of rate of adsorption is adequate to evaluate the samples prepared in this study.

Point 12: L. 358: What humidity? Isotherm for 333.15 K could be provided and compared with 298.15 K. 

Response 12: Thank you for the question. The humidity set in the cycling stability test was at 80% RH and has been stated in Line 402.

333.15 K is the temperature set in our oven only to dry the samples before proceeding with any tests. Sorption isotherm for this temperature is not possible since our adsorption set up does not work at temperature above 298.15 K. The RH and temperature for this experiment were set to imitate the real-life situation in which 80% RH and 298.15 K were the average RH and temperature respectively.

Point 13: L. 384: That happens also at silica gel.

Response 13: Thank you for the feedback. The said phenomenon does happen in silica gel. However, silica gel required higher temperature for desorption (approximately 373.15 – 563.15 K) which may be a disadvantage since it will need more energy and cannot be powered by renewable energy. As included in the manuscript (Line 458) the newly synthesised samples can be regenerated at temperature as low as 318.15 K in which can be powered by renewable energy such as solar energy. This initiative is important in building a better and greener environment.

Point 14: Fig. 11: A comparison with silica gel would be of value. Can this experiment be added?

Response 14: Thank you for the suggestion. Unfortunately, the original set up for this specific test is no longer available, thus we cannot run more adsorption test for comparisons. However, we focus our study on the effect of GO on the adsorption capacity of the copolymer and from Figure 13, we can see that the addition of GO indeed improve the adsorption capacity at all RH%. We believe that these results are adequate and have met our objectives.

Please refer the attachment. Thank you.

Round 2

Reviewer 1 Report

1. The AA structure in Figure 2 was wrong and not corrected.

2. The NMR detection condition should be elucidated in Part of 2. materials and methods. The H peak of CH3  in Figure 6a was inappropriate, which should be a single peak. And water peak was also denoted  inappropriately in Figure 6b.

3. Some format errors should be checked and revised carefully, for example, "2.1   .Materials" 

Author Response

Response to Reviewer 1 Comments

Point 1: The AA structure in Figure 2 was wrong and not corrected.

Response 1: Thank you for the comment. Amendment has been made.

Point 2: The NMR detection condition should be elucidated in Part of 2. materials and methods. The H peak of CH3 in Figure 6a was inappropriate, which should be a single peak. And water peak was also denoted inappropriately in Figure 6b.

Response 2: Thank you for the comment. NMR detection condition has been elucidated in Part 2. The H peak assigned for CH3 was based on the references which have shown triplet peak at the similar ppm. (Ref: https://doi.org/10.1007/s43938-021-00005-8, https://doi.org/10.1007/s11814-014-0031-5, and https://doi.org/10.3390/biophysica2040037). The label for water peak has been corrected.

Point 3: Some format errors should be checked and revised carefully, for example, "2.1   .Materials" 

Response 3: Thank you for the comment. Amendment has been done.

Reviewer 2 Report

Thank you for ammendments.

Author Response

Thank you for the feedback.